# Atmospheric Muon Flux Measurement near Earth's Equatorial Line

Cristian Borja *,†, Carlos Ávila †, Gerardo Roque  and Manuel Sánchez *

Physics Department, Universidad de los Andes, Bogotá 111711, Colombia
* Correspondence: cm.borja10@uniandes.edu.co (C.B.); mf.sanchez17@uniandes.edu.co (M.S.)
† These authors contributed equally to this work.

**Abstract:** We report measurements of muon flux over the sky of the city of Bogotá at $4°35'56''$ north latitude, $74°04'51''$ west longitude, and an altitude of 2657 m above sea level, carried out with a hodoscope composed of four stations of plastic scintillators located equidistant over a distance of 4.8 m. Measurements were taken at different zenith ($\theta$) angles within the range $1.5° \leq \theta \leq 90°$, the muon flux data is statistically consistent with a $cos^2\theta$ dependence, with a $\chi^2$ per degree of freedom near unity. If instead, we fit to a $cos^n\theta$ we obtain $n = 2.145 \pm 0.046$ with a lower $\chi^2$ per degree of freedom. Integrating the muon flux distribution as a function of the zenith angle over the solid angle of the upper Earth's hemisphere allows an estimation of the atmospheric vertical muon rate at the altitude and latitude of Bogota obtaining a value of $255.1 \pm 5.8$ m$^{-2}$s$^{-1}$. This estimate is consistent with an independent direct measurement of the vertical muon flux with all detectors stacked horizontally. These measurements play a key role in the further development of detectors, aimed to perform muon imaging of Monserrate Hill, located in Bogotá, where the detectors will be placed at similar locations to those used in the present study.

**Keywords:** cosmic rays; Monserrate Hill; muography; muon flux; zenith dependence



## 1. Introduction

According to the standard model of particle physics (SM), muons are leptons with a lifetime of $2.2 \times 10^{-6}$ s, very high penetration length [1], the same electrical charge and spin as electrons, but with a mass around 207 times larger. Muons were discovered by Anderson and Neddermeyer in 1936 via studying cosmic rays (CRs) [2]. CRs are remnant energy coming from galactic events, such as interstellar collisions of gasses, solar wind emissions, and supernova explosions. They are composed mainly of high-energy protons (90%), alpha particles (9%), and in a lower proportion, electrons and a small fraction of nuclei heavier than iron (Fe). Secondary particles are produced when the most energetic particles from cosmic rays penetrate the Earth's magnetic shielding. They strike the nuclei of light elements in the upper atmosphere such as nitrogen and oxygen, producing the so-called secondary particle cascades, which continue traveling through the atmosphere until reaching the Earth's surface. Baryons and mesons can interact with each other and decay into other particles as they reach their mean lifetime; charged meson decays produce muons and neutrinos ($K^\pm \rightarrow \mu^\pm + \nu_\mu(\bar{\nu}_\mu)$), and kaon decays produce charged pions that can generate muons through subsequent decays ($\pi^\pm \rightarrow \mu^\pm + \nu_\mu(\bar{\nu}_\mu)$).

The most abundant components of the particle showers arriving at the Earth's surface, which can be detected underground, are neutrinos and muons (see [3]). Muons have a mean lifetime of two orders of magnitude higher than kaons and pions. Given their high mass as compared to electrons, they also have a much lower interaction cross-section with matter. Thus, relativistic effects of length contraction allow muons to travel long distances before decaying or losing all their energy. In consequence, muons can pass through the atmosphere and penetrate hundreds of meters inside mountains and even the

soil, making their detection on Earth possible. Muons can be produced at different depths in the atmosphere and impact the Earth's surface in various directions. Therefore the muon energy spectrum has a dependence on the atmospheric conditions, altitude, and the muon incidence trajectory.

When muons were first detected, everyone confused them to be the $\pi$ mesons predicted by Yukawa in 1935. However, these particles do not interact as strongly with atomic nuclei, just as Yukawa's meson does. The relation between muons and $\pi$ mesons was discovered by Powell and his collaborators in 1947, by using photographic plates exposed to CR in Mount Chacaltaya, in the Bolivian Andes. This experiment showed that a charged $\pi$ meson decays into a muon and a neutral particle, which was later found to be a neutrino [4].

CR studies helped establish the essential properties of muons, such as their mass, spin, and half-life. Nevertheless, these observations on muons from CR, as well as other particle physics experiments, opened the door to new puzzles that still need to be solved: the radius of the proton [5,6], the lepton flavor violation (LFV) [7], the Higgs boson decay to a muon pair [8], the $B_s$ meson decay to a muon pair [9], the anomalous magnetic moment of the muon $(g-2)_\mu$ [10], among others.

Some of the previous anomalies have been studied theoretically in the frame of BSM physics and experimentally in laboratories such as CERN, Brookhaven National Laboratory, Fermilab, PSI, etc. [11–15]. The above are just some examples of the importance of muons, as a tool to understand basic physics questions. However, research on muon physics goes beyond these fundamental issues. Measurements of CR muon flux are also of great interest in particle physics since they are related to experiments that require very low background events (indirect or direct dark matter research, double beta decay detection, proton decay measurements, among others) [16]. Muons represent an important source of background. Thus, if muon flux is not effectively suppressed, rare events related to the neutrino production or hypothetical dark matter candidates cannot be detected [17]. This is the reason why CR muons are used as calibration sources for experiments such as OPERA [18], ANTARES [19] and the Super Kamiokande [20], which investigate neutrino oscillations and flux [21].

The extensive study of muons produced due to the interaction of cosmic rays with the atmosphere from theoretical and experimental points of view has made it possible to use these particles for the development of non-invasive techniques, such as muon radiography, muon metrology, and muon tomography, to image various targets in different size scales.

In this paper, we report measurements of muon flux at $4°35'56''$ north latitude, $74°04'51''$ west longitude, and an altitude of 2657 m above sea level, considering its dependence on the zenith angle. These measurements constitute the first step in imaging muons at Monserrate Hill in Bogotá and future other places at high altitudes throughout Colombia. The muon flux data at low latitudes could complement software such as CORSIKA or FLUKA [22,23] which aim to simulate atmospheric muon production arriving at the Earth's surface. In addition, muon flux measurements as a function of zenith angle are important to understand how muons are generated through the interaction of cosmic rays with the atmosphere. Muons produced at very high zenith angles are usually the tool for imaging geological structures such as mountains or volcanoes; a knowledge of the quasi-horizontal muons is therefore important for such types of experiments. Section 2 provides a review of the most relevant investigations in muography and its applications in different areas such as the imaging of volcanoes and seismic faults. The limitations and difficulties that detection devices have to surpass are discussed, as well as how background sources can affect the muon flux measurements. In Section 3, we explain the physical principles that describe muon interaction with matter and how muography is possible. Section 4 is focused on technical details about our setup, triggering system, and data processing. Finally, Section 5 reports the results of the first measurements of muon flux over the sky of Bogotá.

## 2. Muography Applications

In the 1950s, E.P. George used a Geiger counter to determine the density of an overburden of a tunnel based on muon flux attenuation measurements [24]. Even without directional information for incident muons, the device was capable of detecting flux changes through the rock above the tunnel. This is the first precedent of muon imaging, with the term muography used thereafter. In subsequent years muography and muon tomography have been widely proposed in different areas such as particle physics, archaeology, and geosciences.

In 1970, the first application of muography to archaeology was published. Alvarez et al. suspected the existence of hidden chambers in Kephren's pyramid because two of the Giza pyramids (Keops and Sneferu) already had known secret chambers [25]. The detector employed was composed of a series of spark chambers, which allowed obtaining directional information of the incident muons. The detector was placed in the lower region of Kephren's pyramid and measured muons for several months. Alvarez et al. reported that muon flux through the pyramid was not compatible with the existence of hidden chambers. After 50 years of developments in muography, in 2017 the ScanPyramids project determined the existence of hidden cavities in the great Keops pyramid, employing three different types of detectors: nuclear emulsions, scintillating hodoscopes, and gas detectors [26]. Currently there is an ongoing investigation on the development of a WLS-fiber detector system with high sensitivity capable of a resolution of $\sim$0.8 mm for the muon-detector interaction point. The project aims for a complete tomographic scanning of the Great Pyramid [27].

Not only the Giza pyramids have been studied with muons. In Mount Echia, atmospheric muon detection has been employed to obtain images of well-known underground cavities. The purpose was to evaluate the resolution and performance of the detection system and demonstrate the technique's potential [28]. Gomez et al. performed simulation studies on the Greek Kastas Amfipoli Macedonian Tumulus to evaluate muon tomography capability upon this structure [29]. On the other hand, the Imashirozuka burial mound, built in Japan after an earthquake in 1596, has been investigated recently with muography by Tanaka et al. [30] with the purpose of studying evidence of the historic earthquake. Moreover, recent investigations on muon tomography for archaeological structures scanning have been proposed in Mexico to study the Sun Pyramid in Teotihuacan [31,32].

Even when muography applications in archaeology have shown the technique's potential, muography is most commonly used in volcanology studies. The high volcanic activity in Japan aroused the interest in the use of muography as a monitoring tool: In 1995, Nagamine et al. published measurements of the muon flux attenuation through Mount Tsukuba in Japan, with the use of cosmic muons detected by a three-fold plastic scintillator telescope positioned at 2.0 km from the mountain [33]. They proposed the use of muons to image volcanoes and observe changes in their inner structure, as this is known to be directly related to their activity level. Unlike archaeological bodies, when studying volcanoes, the high rate of vertical muons cannot be used, mainly because of the difficulties to operate a detection system near the caldera. To map these structures, it is necessary to measure incident muons at large zenith angles. Even when the horizontal muons rate is lower, Nagamine et al. accomplished to probe the upper region of Mount Tsukuba by measuring quasi-horizontal muons. The results allowed to distinguish density changes in the observed region [33].

In 2006, Tanaka et al. obtained a radiographic image of the Mt. Asama crater using emulsion detectors [34]. The setup was placed underground inside a vault to reduce the background related to other charged particles. The results allowed the detection of a dense region within the crater, which corresponds to a lava mound formed during a recent eruption (2004). Years after, in 2009, a volcanic eruption in Mt. Asama was recorded in a muography. Tanaka et al. were capable of measuring a mass loss of $31 \times 10^6$ kg due to the mass ejections during the eruption using a scintillator-based detector [35]. In 2013, using a similar scintillation detection system, Tanaka et al. reported for the first time a muographic observation of magma dynamics in the conduit of the Satsuma–Iwojima volcano. They

observed ascending and descending magma columns when comparing time-sequential muographic images [36].

One of the motivations for using muons is the increased spatial resolution that is accomplished compared to traditional gravimetry methods [37,38]. The combination of those methods have allowed the tomographic reconstruction of the density structure of volcanoes and the detection of magmatic bodies. In Usu, Japan, Nishiyama et al. built a 3D model of the density structure of the Mt. Showa-Shinzan lava dome by combining gravimetry data and muographic measurements made with a multilayered emulsion detector [39].

The DIAPHANE project is the most extensive collaboration focused on the development and application of muon tomography for volcanology studies and monitoring [40]. Most of its measurements have been made on the La Soufriere of Guadeloupe lava dome. Within the scope of this project, Lesparre et al. obtained the first density-muon radiograph of the volcano by measuring muons with scintillator matrices [41]. Jourde et al. studied density changes caused by the hydrothermal activity inside the volcano using muographic images taken from three different detectors located around the structure [42]. Later, Rosas-Carbajal et al. built a tomographic model of the lava dome using joint measurements of muon flux and gravimetry data [43]. More recently, LeGonidec et al. reported that including seismic data enhances the resolution for monitoring hydrothermal activity, which represents a relevant advance towards the prediction of volcanic activity with a time precision of hours to days [44].

Also worth mentioning is a large-scale experiment such as the Sakurajima Muographic Observatory (SMO). The SMO aims to monitor the active volcano Sakurajima in Kyushu, Japan. The measurements employ an improved version of the classical multiwire proportional chambers (MWPCs) redesigned for low gas consumption and enabling the detectors for open field measurements [45]. In 2018, Oláh et al. obtained the first high-definition and low-noise muography of the internal structure of the Sakurajima erupting volcano [46]. Later, time-sequential muographic images allowed observing the formation of a magmatic plug after eruption activity [47]. Post-eruptive measurements have also demonstrated the high sensitivity of the technique for monitoring purposes and its potential for the early detection of volcanic hazards [48].

In Italy, various collaborations have used muons to map different volcanic bodies. The first measurements on the southern crater of the Etna volcano were conducted by Carbone et al. in 2010 at an altitude of $\sim$3.0 km, with an elevation of $\sim$250 m [49]. The experiment allowed testing a pixel scintillation detector under open field conditions and also understanding background noise arising from fake muon detection. The authors discuss that the probability of accidental detection lowers when three or more simultaneous coincidences occur. Furthermore, the trajectory discrimination across the different pixel detectors can improve the noise reduction. After three years, the authors obtained the first muography of the Etna volcano using a model to estimate the noise and enhance the flux estimation [49]. Later, the Muon Radiography of Vesuvius (MURAVES) project aimed to study the Vesuvius crater and infer potential eruption pathways using an improved version of the Muon Ray (MU-RAY) detector [50]. In 2013, the Tomography with Atmospheric Muons from Volcanoes (TOMUVOL) and MURAVES collaboration attempted to make a joint measurement of the transmittance of the inner structure of the Puy de Dôme in France [51,52]. In 2018, Tioukov et al. reported the first experiment with nuclear emulsion muography at the Stromboli volcano, allowing the measurement of the density structure beneath the volcano crater [53]. More recently, the Muography of Etna Volcano (MEV) project focuses on developing detectors intended for studying volcanoes [54] in collaboration with geoscientists, engineers, and physicists. The special requirements for this kind of open field application are transportability, good spatial and time resolution, and low electric power consumption, as usually electric power for these applications is obtained by solar panels.

In Colombia, various studies have been made to choose geological bodies that can potentially be used as targets to test muon imaging techniques. In particular, preliminary

studies on the design and construction of a muon telescope have been performed for the future imaging of the volcano Machín in Tolima, Colombia [55]. Recently in Bogotá, simulation studies of attenuation flux through Monserrate Hill have been performed. Preliminary estimates of muon flux dependence with zenith and azimuth angles have been obtained [56]. Monserrate Hill is located on the western edge of the Bogotá mountain range. It originated from Bogotá's fault, primarily composed of layers of sandstone and clay, with the presence of some layers of phosphorite [56,57]. The highest point of the hill is located at 3176 m above mean sea level and its GPS coordinates are 4°36′18″ north latitude 74°03′19″ west longitude, and it has a prominence of 550 m above Bogotá's average altitude. The study of Monserrate becomes important since it is of great geological interest to unveil its internal structure, expanding the knowledge about the topography of the city [56]. Our university is located near Monserrate Hill (~1.2 km), providing a strategic place to mount the detectors used in the present study.

For further details and experiments overviews, we recommend previous review articles: Bonechi et al. [58] focus on background noise reduction in muography; Rhodes [59] reviews muon tomography applications related to non-proliferation; Procureur [60] offers an overview of muon imaging applications at different size scales and emphasize on recent advances in the last 15 years; Bonomi et al. [61] presents some additional applications of cosmic-ray muon detection; Tanaka and Oláh [62] explore the evolution of some research networks focused on muography; Leone et al. [63] summarize some studies on muography applied to volcanoes; and finally Kaiser [64] discusses commercial aspects on muon detectors and the future direction of muography applications. We also suggest the book by Oláh et al. [65], in which the principles of muography, different approaches for muographic imaging techniques and the latest technological developments in muography are explained.

## 3. Physical Principles of Muography

The choice of the radiation source to image a target depends on its properties (size, and composition). With X-rays, one can image objects in the μm to mm scale [66]. However, the imaging of thicker objects requires a higher energy radiation source. There are different kinds of naturally-produced radiation that can be used for these purposes: beta and gamma radiation, neutrons, muons, or even alpha particles, which are useful for geological and mineral studies [67]. Atmospheric muons are particularly attractive. Although their intensity is naturally low, they are highly penetrating and, at the same time, they are a free and unlimited source of radiation, without harmful health effects.

Muons interact with matter and lose energy through a series of radiative processes: Bremsstrahlung radiation, production of $e^+e^-$ pairs, and photonuclear interactions. The mean amount of energy loss can be expressed as a function of the amount of matter traversed (in MeV cm$^2$ g$^{-1}$ units) [3]:

$$\frac{-dE}{dx} = a(E) + b(E)E, \tag{1}$$

where $b(E)$ considers pair production, Bremsstrahlung, and photonuclear contributions, and $a(E)$ describes ionization losses, which depend on the target characteristics such as its density and composition. In some muon imaging applications, this magnitude is referenced as "stopping power", where $a(E)$ is the electronic stopping power and $b(E)$ is the energy-scaled contribution.

The mean rate of energy loss for moderately relativistic charged heavy particles is described by the Bethe-Bloch equation [3] (This equation is valid in the range $0 \leq \beta\gamma \leq 1000$):

$$\left\langle \frac{-dE}{dx} \right\rangle = Kz^2 \frac{Z}{A} \frac{1}{\beta^2} \left[ \frac{1}{2} ln \frac{2m_e c^2 \beta^2 \gamma^2 W_{max}}{I^2} - \beta^2 - \frac{\delta(\beta\gamma)}{2} \right], \tag{2}$$

where $Z$ and $A$ are atomic and mass numbers, $z$ is the muon charge, $K$ is a constant ($K \approx 0.307 \frac{\text{MeV cm}^2}{\text{mol}}$), $W_{max}$ corresponds to the maximum energy received by an electron

through a collision, $I$ is the mean excitation energy and $\delta$ is the density effect correction to ionization energy loss, $\beta$ is the particle velocity over $c$, and $\gamma$ is the Lorentz factor.

Muons have a significantly higher probability of surviving the atmospheric interaction, as compared to other secondary particles, mainly because of the contribution of two effects: the already mentioned longer lifetime, and their large mass compared to electrons. Bremsstrahlung losses are proportional to $1/m^2$, which results in a loss about 40,000 times smaller for muons than for electrons.

### 3.1. Transmission Muography

As a consequence of the muon energy loss, the probability of detecting a muon that travels across a certain amount of matter is reduces when its energy loss approaches its initial incident energy. This phenomenon leads to a quantity called opacity: A measure of the muon flux attenuation caused by the target, compared to the open sky muon flux. The opacity allows to infer the average density of an object when considering its geometric features. This technique is possible because of the wide energy distribution of cosmic muons, as some of them can travel through several hundred meters of rock. In consequence, opacity can still be measured for very thick objects. Thicker targets require a larger imaging time due to the naturally low-intensity muon rate. This effect reduces the range of applications to those that are not time-critical (e.g., volcanoes, mountains, archaeological structures, etc.). Experimentally, the measurement of transmitted muon flux through a target is carried out by pointing the muon detector towards different regions on the object and measuring muon flux during a time window proportional to the target dimensions and the detector's spatial resolution. Usually, the time required to obtain a muography may vary from weeks to months, as discussed in Section 2.

### 3.2. Muon Flux Zenith Angle Dependence

Although muon flux is stable, it can be affected by several effects such as its dependence on zenith angle and geographical location [68]; the west-east asymmetry [69], the geomagnetic activity [70–72], changes of temperature and atmospheric pressure [73–75], among others.

Muon intensity follows a cosine distribution of the zenith angle $\theta$ as:

$$I(\theta) = I_0 cos^n(\theta), \tag{3}$$

with $n$ value depending on the atmospheric conditions and geographical location. Experimentally, $n$ value is $n \approx 2$ [76,77]. However, the $cos^2(\theta)$ law fails for high energy muons [78,79].

The Earth's geomagnetic field plays an important role in the number of primary particles that arrive and interact in the atmosphere. The main effect of the geomagnetic field is the shielding or cutoff in energy for a given particle to enter the atmosphere. This cutoff is given by the contribution of various effects related to altitude, longitude, and east-west asymmetry. The different measurements of muon flux [76,77,80–82] suggest that its intensity decreases with geomagnetic latitude, close to the equator [83]. Another effect involved is the variation of flux intensity due to the asymmetry of the geomagnetic axis respect to the Earth's rotation axis.

Additionally, the flux coming from the east and west directions present a difference in energy of up to 100 GeV. This difference is more appreciable at higher altitudes because the dependence on the zenith angle screens it at sea level. The understanding of the atmospheric and environmental effects over muon flux intensity represents an advantage for the performance of attenuation muography; moreover, it enables the implementation of corrections to data [49]. On the other hand, the time required to image a target depends on aspects such as the detector size, angular resolution, the detector position relative to the target, atmospheric depth, and the required precision of density, which depends on the application. Leone et al. discuss relevant features for obtaining muon images within the optimal time scales to accomplish early warnings for volcanic hazards [63].

Analyzing data obtained by the MACRO experiment for underground muon detection [84], Ambrosio et al. studied the effects of the moon transit shadow and the solar transit effect on muon flux. Finally, differences in muon counts during the day and night suggest that the change in geomagnetic configuration due to solar winds also affects the muon flux [85].

## 4. Materials and Methods

The muon telescope used for the present study is composed of four stations of scintillator plastic detectors, equidistantly separated within a range of 4.8 m. By registering straight-through muons going through all four detector stations, a high reduction of fake muon backgrounds is achieved. The detectors are assembled within lateral metallic frames that are then mechanically attached to retractable solid aluminum rails with a maximum length of 5 m.

The longitudinal detector position can be adjusted to have different solid angle coverage. However, for the measurements reported here, a fixed distance of 1.6 m was used between adjacent detector stations. The rails are mounted on a heavy stainless steel base that provides mechanical stability to the system and allows angular repositioning, in both azimuth and zenith angles, with a precision of 0.5°. There is an additional adjustable tripod coupled to the lower end of the telescope for further mechanical stability and to minimize vibrations in the system (see Figure 1).

There is an average of about 1000 mm of rainfall precipitation per year around the Monserrate Hill location, winds could reach speeds of up to 40 km/h on a few occasions and have a yearly average speed of about 13 km/h. The daily temperature varies within the range of $[5°, 25°]$ Celsius [86,87]. These climatic conditions require special attention because they could degrade the detector performance. For example, optical coupling with epoxy material as the two coupled surfaces could have a different thermal expansion, so the daily temperature gradient of 20 degrees Celsius could degrade such couplings. Moreover, the muon telescope is more sensitive to wind variations when pointing towards small zenith angles, producing higher systematic uncertainties. To protect the detectors from different climate conditions, such as rain, day-night temperature variation, sunlight, and wind, the detectors are wrapped with styrofoam and encapsulated inside a plastic box which is in turn lined with a thick black plastic foil.

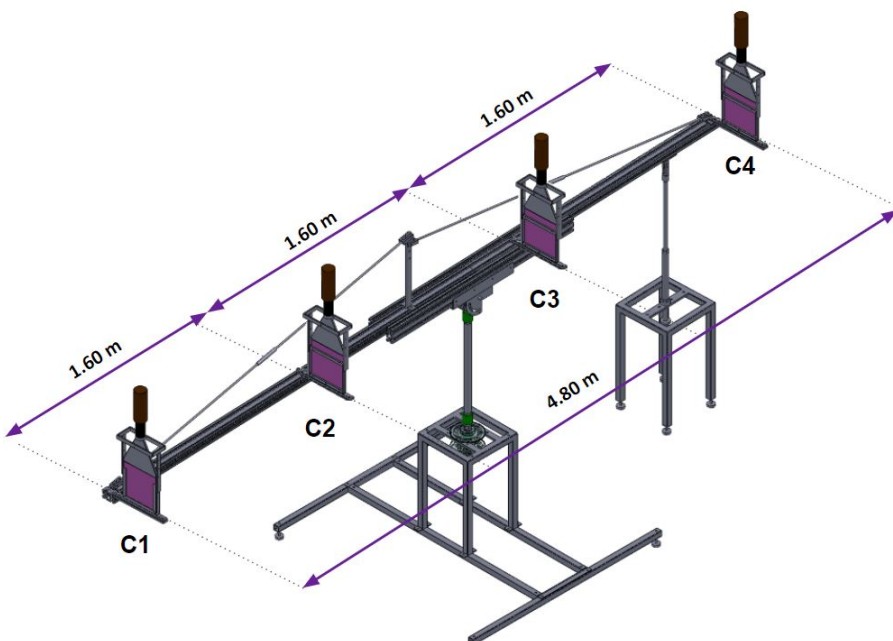

**Figure 1.** Layout of the telescope structure. The detectors are located equally spaced.

### 4.1. Scintillation Detectors

Each detector consists of a polished 25 cm × 25 cm × 1 cm Polyvinyltoluene scintillator block, reference Saint-Gobain BC-416, which emits fluorescence photons (with a peak spectral emission of $\lambda = 434$ nm) due to the molecular excitation produced by incident charged particles.

The block is attached, with a thin epoxy layer, to a fish-tail light guide (LG) made of acrylic plastic (PMMA), reference Saint-Gobain BC-802, with a 95% transmission for wavelengths of 425 nm. The other extreme of the LG is coupled to a photomultiplier tube (PMT). Since the PMT Borosilicate glass window and the LG have different thermal expansion coefficients, preliminary tests indicated that optical grease achieved a much better coupling than epoxy for the temperature gradients to that the system was exposed. The PMT used is a Hamamatsu H6410 with a peak spectral response at the wavelength of 420 nm, providing a good match to the emission spectrum of the scintillator, and with a rise time of 2.5 ns. Incident photons emitted by the scintillator generate, through the Photoelectric effect, an initial electron cascade that gets further amplified by the electrons being accelerated by the PMT high voltage and hitting the different dynodes within the PMT. The final electron cascade arriving at the PMT anode allows the detection of the interaction of a muon with the scintillator block as an electric signal. A Low-Level Threshold (LLT) is defined, as described later in the text, to discriminate PMT signals and decide whether or not a signal is considered an event above the background noise.

Figure 2 indicates the $\Delta\theta$ and $\Delta\phi$ angle coverage of the trigger detectors ($C_1$ and $C_4$) when the telescope is oriented at a zenith angle $\theta$. The length of the detectors is $L = 0.25$ m and the distance between the two farthest detectors is $D = 4.84$ m. Therefore, the detector has an angular resolution of about $3.0°$. On the other hand, the maximum $\Delta\theta$ covered is $\Delta\theta = \tan^{-1}(L/D) \approx L/D$ and the maximum $\Delta\phi$ is $\Delta\phi = \tan^{-1}(L/D\sin\theta) \approx L/D\sin\theta$. Thus, the solid angle, $\Delta\Omega$, coverage can be calculated as:

$$\Delta\Omega = \int_{\theta}^{\theta+\Delta\theta} \int_{-\Delta\phi}^{\Delta\phi} sin\theta d\theta d\phi = \int_{\theta}^{\theta+\Delta\theta} 2\frac{L}{D} d\theta = 2\frac{L^2}{D^2} = 0.00534 \; sr. \tag{4}$$

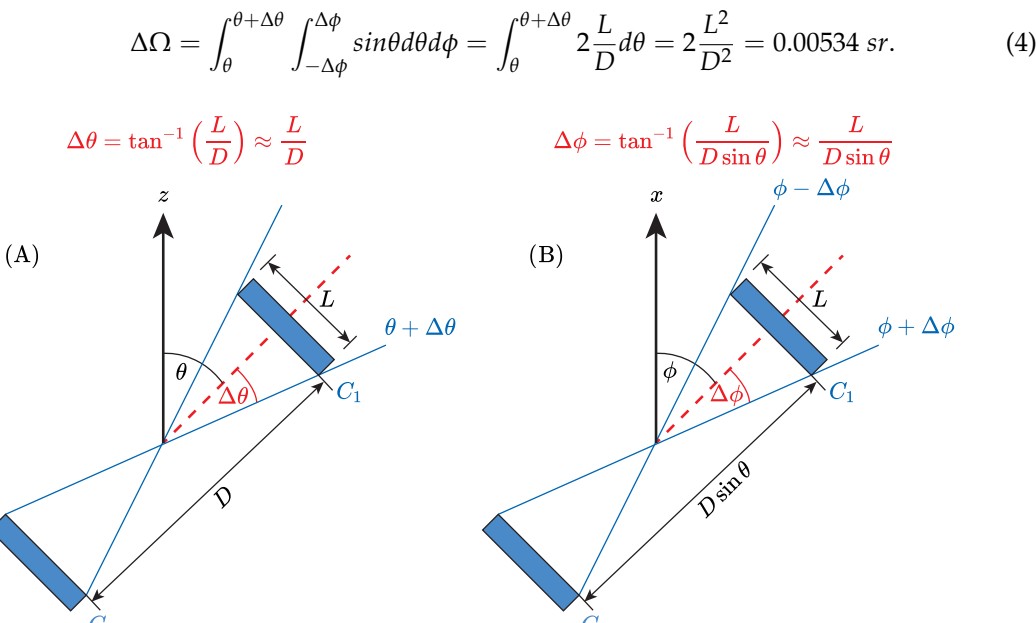

**Figure 2.** (**A**) Longitudinal view and (**B**) lateral view of the two farthest detectors of the telescope.

### 4.2. Triggering System and Event Discrimination

Since for the present study we only have four detectors to be read out, we use standard NIM and CAMAC electronics which are suitable for a low number of independent readout channels. As shown in Figure 3, the analog signal from each PMT, which has a negative amplitude, is fed into a FANIN/FANOUT module to duplicate it and distribute it to the digitizing modules, one duplicate of the signal is then entered to a discriminator

module, while the second duplicate is properly delayed to arrive within the time integration window of the charge digital conversion module (QDC), in our case a 90 ns delay has been determined to be optimum.

In the discriminator module, a Low-Level Threshold (LLT) of $-200$ mV was set for each detector to decide whether or not a signal is considered an event above the background noise. Once the amplitude of the analog signal exceeds the LLT, the discriminator module produces a NIM digital signal, indicating the existence of an event generating enough charge to be consistent with a real muon hitting the detector. The NIM signal has a rectangular shape with a fixed amplitude of $-1.5$ V and a width that is set by the user, in our case 20 ns is sufficient for comparison with signals from the other detectors.

The two farthest detectors (*C*1 and *C*4 in Figure 3) are chosen as the trigger detectors because they define the solid angle coverage of the whole system. A trigger to store data is generated when digital discriminator signals from the two trigger detectors arrive in coincidence. This two-fold coincidence is performed in the "AND" module indicated in the figure. If the two input NIM pulses from the trigger detectors are overlapped for at least 1 ns, a coincidence signal is produced. The output of the coincidence module is made wide enough (100 ns) to serve as the time window for the integration of charge in the QDC module. The leading edge of the trigger signal is also used to start an analog Binary Coded Decimal counter (BCD) to count every event detected by the whole system. Finally, the leading edge of the trigger pulse acts as the common start of the time digital converter module (TDC).

Output signals from the discriminator of each detector are delayed by 64 ns and entered into different channels of the TDC module to stop four independent time counters, one for each detector. In this way, digitized timing information from each detector can be registered. Both QCD and TDC modules are installed in the same CAMAC crate. The CAMAC controller uses the common bus of the crate to send control commands to the modules and also acts as an interface with a PC through a USB connection. Commands to the CAMAC controller are managed on the PC with a LABVIEW software code.

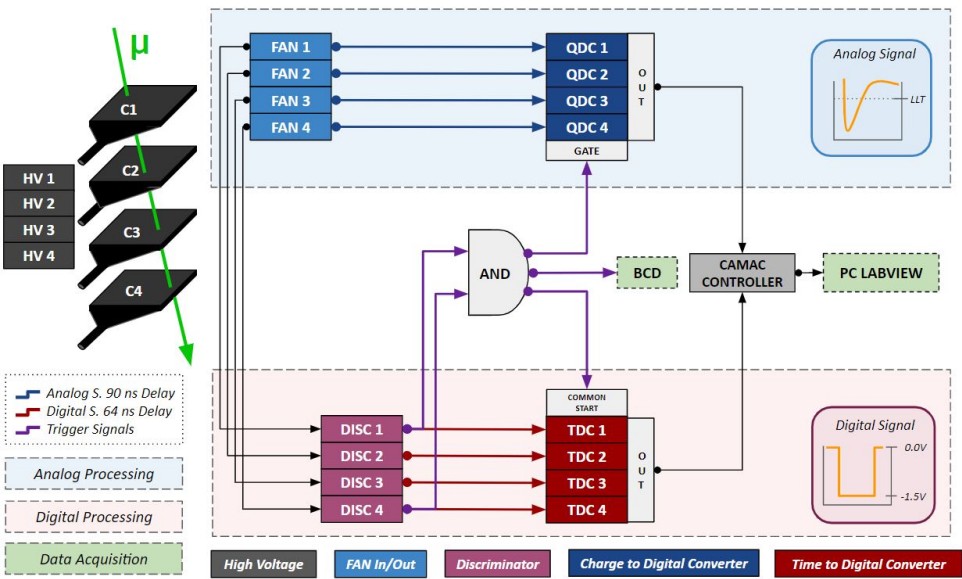

**Figure 3.** Electronic diagram for the UNIANDES muon telescope. The diagram shows the processing of analog and digital signals and summarizes the signal processing phase.

Each digitized timing unit in the TDC module corresponds to a nominal time of 25 ps. The maximum range of digital time counts is 4096 for each TDC channel, having in total an active time window of about 102 ns for each measurement. The TDC module is set in a common-start mode and measures the time difference between the stop signal of each channel with respect to the common initializing signal. The trigger coincidence signal

activates the start mode of the TDC counter. After an initial dead time of 20 ns, the time window opens, and each TDC channel receives and records the time stamp of the leading edge of the NIM pulses coming from each detector discriminator. Finally, the CAMAC crate controller, after receiving a copy of the trigger pulse, gathers the CAMAC TDC and QDC information and sends it to a PC, where the four detector time-stamps are read by the LABVIEW program and recorded for further offline analysis. To determine the calibration between Time digital counts in the TDC unit and the real timing in ns, we used a pulse generator with different delays, ranging from 5 ns to 100 ns in 5 ns steps, for each channel of TDC module. There is a linear correlation in most of the active TDC range, for each channel, with some small variations at the beginning and end of the range. Proper delays were set up to have the timing range of each TDC channel within the linear region.

It is also possible to perform forward/backward incident muon discrimination without a very high time resolution. The possibility of making this distinction is an advantage when the direction of incidence of the muon becomes relevant. Generally, the upwards muon flux is assumed to be zero, thus pointing vertically at a hodoscope and measuring vertical muons is different than when it is used to measure large zenith-angle muons. In the first case, it can be assumed that all muons came in from the top. However, if we point at a large zenith-angle, muons can pass in both directions with similar probability. With the two trigger detectors separated by 4.8 m, a muon Time-of-flight (ToF) of 16 ns between them is expected. By delaying by 16 ns the signal of the first detector hit by a muon traveling downwards from the sky, it can be assured that both incoming signals from the trigger detectors arrived fully synchronized to the coincidence unit. However, if the muon travels upwards, the time difference between the two trigger detectors would be ToF+Delay, which is 32 ns. Since the pulse width (PW) for each detector discriminator is 20 ns, the pulses generated by the two trigger detectors for the upwards-traveling muons will be out of time without making a coincidence.

Some fake detections can be triggered by uncorrelated events (e.g., a muon strikes one of the trigger detectors, and electronic noise in the second scintillator detector surpass the threshold level). For this reason, once the data are collected, an offline four-fold coincidence is required. In this way, we only consider events measured by all the detectors and reduce the number of fake detections.

The data stored in the PC come from the TDC and QDC modules. The LabVIEW data acquisition system (DAQ) reads this information with a dead-time of 1 ms between acquisitions, sending a signal at the end of the acquisition to reset the CAMAC modules. The software records the number of events detected. To correct for the lost events due to dead times, we use the information from the BCD counter, which has the total number of trigger events produced. This allows calculating the DAQ detection efficiency as $\epsilon_{DAQ} = C_S/C_T$, where $C_S$ is the number of counts recorded by the DAQ system and $C_T$ is the total counts recorded by the BCD counter. We found that this efficiency varies between 98% to 99% for the different zenith angles used for taking measurements with the telescope. When the detectors are taken out of the telescope and piled up horizontally one on top of the other, to measure the muon vertical flux, the DAQ efficiency is 84% due to the higher event rate processed by the system.

In an offline analysis, additional criteria are implemented to consider an event as a real muon detection. After imposing the four-fold coincidence, a QDC cutoff value (QDC CV) of 200 DAQ units is defined to discard low-energy events related to electronic noise in the PMTs. We estimate this QDC software threshold by looking at the intersection of the pedestal noise and the lower tail of the muon signal in the QDC as indicated in Figure 4.

*4.3. Detector Performances*

A high-voltage (HV) plateau curve was obtained for all four detectors using vertical muons, with all detectors stacked horizontally and with an initial low discriminator value of −100 mV. The plateau curve for a specific detector $C_i$ was obtained by requiring a coincidence with the other three detectors that were operated initially at a HV of 2100 V.

A scan on the HV of the $C_i$ detector was performed and the distribution of events of the $C_i$ detector in coincidence with the other three detectors, normalized by the counts of the other three detectors in coincidence, was plotted as a function of its HV (See left panel of Figure 5 for the case of $C_2$). We then defined as the operating HV for this $C_i$ detector a value of 100 V above the turn-on curve observed. We iterated this procedure for all four detectors.

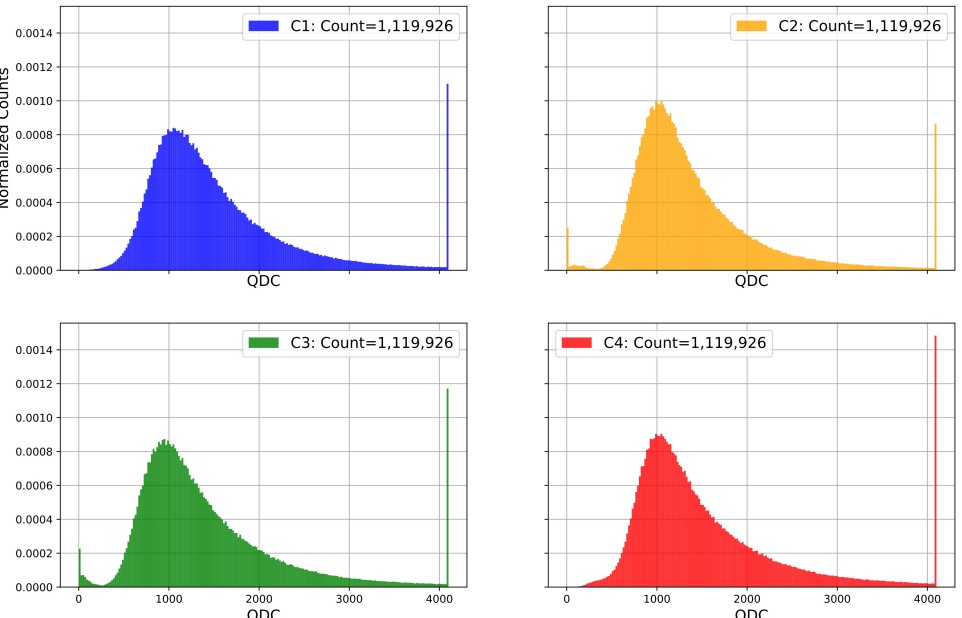

**Figure 4.** QDC histograms for each of the four detectors. The small low tail to the left in the non-trigger detectors corresponds to noise. The spikes on the right sides correspond to overflow due to sporadic high-energy events.

Once we found the operating HV, we also performed a scan on the discriminator thresholds for each detector. We changed the discriminator thresholds in steps of $-50$ mV and plotted the number of four-fold coincidences, normalized by the coincidences of the other three detectors, as a function of the discriminator threshold for detector $C_i$. We set the LLT value 100 mV higher than the value at which the curve started to drop down (see Right panel of Figure 5 for the case of detector $C_2$). The optimal LLT found was $-200$ mV, the same value was set up for all four detectors. The optimal operating HV was 2100 V for detectors $C_1$, $C_2$ and $C_4$, while the HV for detector $C_3$ was 2300 V.

The QDC distributions obtained at the beginning of the data taking, with these operating parameters and triggering on the coincidence $C_1 \cdot C_4$ are shown in Figure 4, where we observe a full signal Landau peak for each of the four detectors, with minimal pedestal noise and with a small percentage of overflow. These QDC curves demonstrate that the detectors are operated very efficiently since the full signal distribution is observed.

The efficiency of detector $C_i$, namely $\epsilon_i$, is defined as the ratio of events measured by the four-fold coincidence over the number of events registered by the coincidence of the other three detectors, as indicated by Equation (5).

$$\epsilon_i = \frac{C_i \cdot C_j \cdot C_k \cdot C_l}{C_j \cdot C_k \cdot C_l} \quad i,j,k,l = 1,..,4; \quad i \neq j \neq k \neq l. \tag{5}$$

The measurement is executed with all detectors aligned horizontally one on top of the other, to have the same solid angle coverage, and requiring events with a QDC above a cutoff value (CV) of 200 units. A summary of the final operating parameters and efficiencies for the detectors are listed in Table 1. The efficiency was measured at the beginning, middle and end period of data taking and proved to be very stable with time. However, a small

deterioration in efficiency was observed in detectors $C_2$ and $C_3$ after the first assembly of the hodoscope for initial tests.

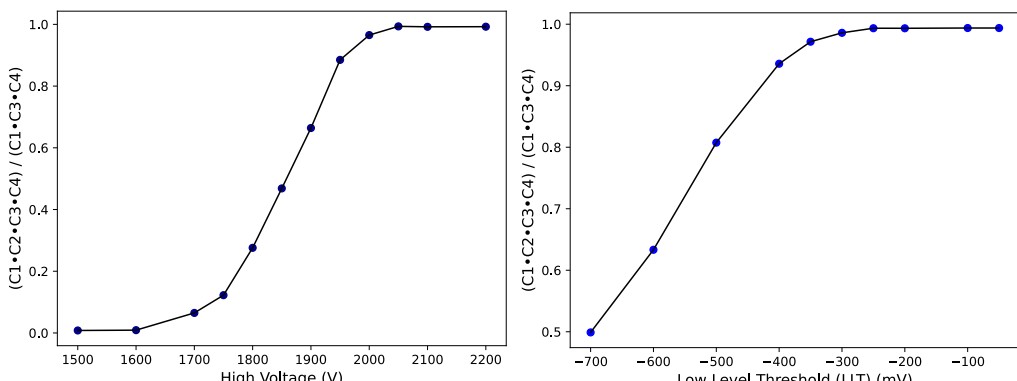

**Figure 5.** Detector characterization for detector $C_2$, the same procedure was followed to find the values of the other detectors, obtaining similar type of curves. (**Left**): Plateau curve for the determination of the optimal HV. (**Right**): Plateau curve for the determination of the optimal LLT.

**Table 1.** Summary of Detector Operation Parameters.

| Detector | HV (V) | PW (ns) | LLT (mV) | QDC CV (DAQ Units) | Efficiency |
|---|---|---|---|---|---|
| C1 | 2100 | 20 | −200 | 200 | 0.996 |
| C2 | 2100 | 20 | −200 | 200 | 0.983 |
| C3 | 2300 | 20 | −200 | 200 | 0.979 |
| C4 | 2100 | 20 | −200 | 200 | 0.992 |

## 5. Results

The azimuth angle $\phi$ is measured with respect to the north direction, while the zenith angle $\theta$ is measured with respect to the vertical direction. For the data measurement, the telescope was pointing towards the sky and changing the zenith angle from $0°$ to $90°$ in steps of $5°$. The data collection was performed from October 2021 through January 2022 with different time exposures per zenith angle direction. Due to the natural low muon intensity, the data were taken for a minimum of two days for each zenith angle measured and increasing the DAQ times for larger zenith angles. The time duration for the last measurements of 85 and 90 degrees was 8 and 13 days, respectively.

The muon flux for a specific zenith angle is determined by counting the number of events having a 4-fold coincidence, $N_{1-4}$, and by dividing each detector efficiency, by the efficiency of the DAQ system, the area $\Delta A$ of the detectors, the solid angle coverage and by the real-time $\Delta t$ of data taking:

$$\frac{dN}{d\Omega} = \frac{N_{1-4}}{\epsilon_1 \epsilon_2 \epsilon_3 \epsilon_4 \epsilon_{DAQ} \Delta\Omega \Delta t \Delta A}. \tag{6}$$

To be detected by our experimental apparatus, a muon has to traverse 4.8 m of air and four scintillator detectors of 1 cm thickness each, and an extra shielding material for light and rain that we estimate to be about 10 cm thick for all four detectors. Muons with low momentum (below 0.1 GeV) have higher interaction cross sections with matter because they fall below the minimum ionizing energy, having high stopping power and losing their energy quickly after traversing short distances [88]. Therefore, we estimate the threshold muon energy of our experimental apparatus to be about 0.1 GeV. Due to their low mass, electrons lose energy through Bremsstrahlung by a factor of $10^4$ higher than muons. On the other hand, protons with energies below 1 GeV drastically lose their energy after traversing short distances because for these low energies their stopping power has a steep increase [89]. Additionally, at the altitude over sea level for our measurements the muon flux is about two

orders of magnitude higher than electrons and over a factor of 10 higher than protons [3]. Therefore, atmospheric electrons and protons have a negligible background contribution to the measured muon flux. To determine the rate from accidental coincidences, we took an additional measurement for the zenith angle of 90 degrees with a 60 ns delay in one of the trigger detectors, in this way, any trigger observed was not consistent with the time of flight of straight through particles. After requiring the four-fold coincidence offline and determining its rate, this background (less than 1% for low zenith angles) was subtracted from the measured muon flux distribution as a function of zenith angle.

Final flux values with their uncertainties are listed in Table 2. Uncertainties in the detector efficiencies are about 0.5%, which depends on the statistic of vertical muons used for each efficiency measurement. We estimated an uncertainty of 2 cm for the measured distance of the two trigger detectors and 0.2 cm for the measured length of the detectors, these values contribute to an uncertainty of 1.8% in the solid angle determination. The statistical error of the DAQ efficiency is of the order of 0.1% to 0.2% and depends on the BCD and DAQ statistics taken. All these uncertainties together with the statistical uncertainty of the four-fold coincidence counts to obtain the final one sigma error quoted in Table 2. For the four zenith angles smaller than 20 degrees, the poorer mechanical stability of the telescope added additional systematic uncertainties, due to vibrations caused by wind. A few measurements were repeated at the same zenith angle and it was observed that some of them were different by more than two standard deviations. To take into account additional uncertainties due to these vibrations, we took the average value of the measurements at the same zenith angle and added in quadrature to the statistical uncertainty the difference between the central value to the extreme measured value. This is the reason for the higher uncertainties reported for these angles.

**Table 2.** Muon flux measurements for the different zenith angles and the corresponding flux uncertainties.

| $\theta$ (Degrees) | $dN/d\Omega$ ($m^{-2}s^{-1}sr^{-1}$) | $\Delta(dN/d\Omega)$ ($m^{-2}s^{-1}sr^{-1}$) |
|---|---|---|
| 1.5 | 130.25 | 6.89 |
| 5 | 134.07 | 6.12 |
| 10 | 125.97 | 7.26 |
| 15 | 109.11 | 8.56 |
| 20 | 110.53 | 5.09 |
| 25 | 106.05 | 3.84 |
| 30 | 92.89 | 3.29 |
| 35 | 82.97 | 3.31 |
| 40 | 68.98 | 2.75 |
| 45 | 61.33 | 2.46 |
| 50 | 48.85 | 2.22 |
| 55 | 37.67 | 1.84 |
| 60 | 29.28 | 1.61 |
| 65 | 20.60 | 1.36 |
| 70 | 12.97 | 1.16 |
| 75 | 7.94 | 1.06 |
| 80 | 2.37 | 1.04 |
| 85 | 1.45 | 0.90 |
| 90 | 0.72 | 0.85 |

We fit a cosine squared function ($f(\theta) = I_0 cos^2\theta$) to the measured flux distribution as shown in Figure 6. The $\chi^2$ per degree of freedom is 1.04, indicating that the measured distribution is statistically compatible with a $cos^2\theta$ dependence with the zenith angle. We also perform a fit allowing the exponent of the $cos\theta$ distribution to vary freely ($f(\theta) = I_0 cos^n\theta$), as indicated in Figure 7. For this second case, we obtain a lower $\chi^2$ per degree of freedom ($\chi^2/ndf = 0.47$), for an exponent slightly higher than 2 ($n = 2.145 \pm 0.046$). By using

the covariance matrix of the fits we also determine the one sigma uncertainty of each fit function as indicated by a yellow band in the figures.

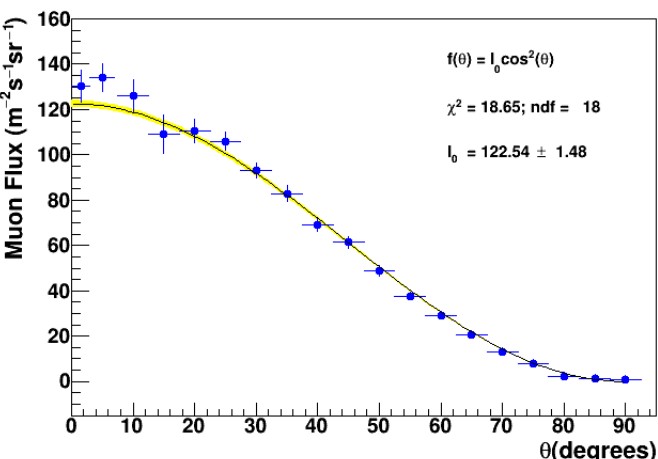

**Figure 6.** Fit of the measured muon flux distribution to a $cos^2\theta$ function. The yellow band shown corresponds to the one sigma uncertainty.

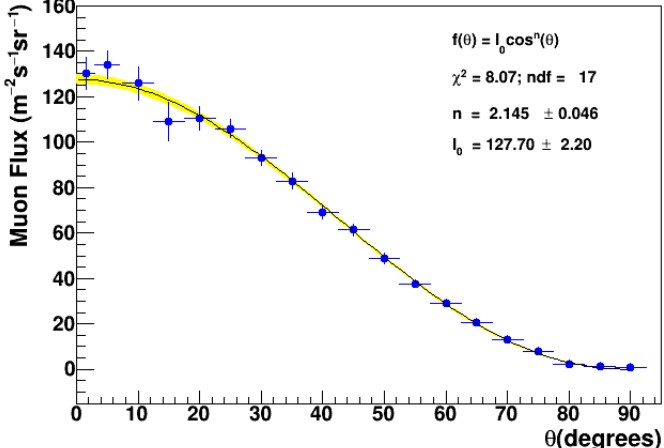

**Figure 7.** Fit of the measured muon flux distribution to a $cos^n\theta$ function. The yellow band shown corresponds to the one sigma uncertainty.

By integrating the measured $I_0 cos^2\theta$ distribution over the solid angle covered by the upper Earth's hemisphere we can obtain the vertical muon flux rate, $I_V$:

$$I_V = \int_0^{2\pi} \int_0^{\pi/2} I_0 cos^2\theta sen\theta d\theta d\phi = \frac{2\pi}{3} I_0. \tag{7}$$

Therefore, $I_V = 256.6 \pm 3.1$ m$^{-2}$s$^{-1}$. If instead, we integrate the $I_0 cos^n\theta$ distribution, with $n = 2.145$ we obtain $I_V = 255.1 \pm 4.4 \pm 3.8$ m$^{-2}$s$^{-1}$, where the first quoted error corresponds to the propagated uncertainty of $I_0$ to $I_V$ and the second corresponds to the propagated uncertainty of $n$ to $I_V$. As observed, both ways to determine the vertical muon flux are fully statistically compatible. We quote as the final number the latest, which includes variations in $I_0$ and $n$.

We performed a second independent measurement of the vertical muon flux ($I_{V,2}$) by piling up all four detectors one on top of the other, triggering the two extreme detectors and requiring a four-fold coincidence. We took data for 24 h. The counts obtained were corrected by DAQ and detector efficiencies, then were divided by the area of the detectors and by the time for data taking, we obtained $I_{V,2} = 261.2 \pm 7.8$ m$^{-2}$s$^{-1}$. This value is

statistically compatible with the previous measurement obtained by integrating the muon flux as a function of the zenith angle.

## 6. Discussion and Conclusions

We have presented a measurement of muon flux as a function of the zenith angle for a latitude very close to Earth's equatorial line and an altitude of 2657 m over sea level. Even though the measured distribution is statistically compatible with a $cos^n\theta$ with $n = 2.0$, the data is better described, with a lower $\chi^2$, for an exponent $n = 2.145$ which differs by 3.2 standard deviations from the value of $n = 2.0$. A comparison of the results from various experiments with the ones presented in this study is shown in Table 3. The $n$ value from the current study is comparable with results from Greisen [80,90], Crookes and Rastin [91], Judge and Nash [76], Sogarwal [92], S. Pal [77] and Pethuraj et al. [83]. However, the value of the vertical integrated flux, $I_0$, is considerably higher than all reported data in Table 3, which is due to the elevated altitude at which measurements were taken.

**Table 3.** Comparison of values of $n$ and integral over flux with other experiments.

| Authors | Latitude (°N) | Altitude (m) | $n$ Value | Vertical Integrated Flux, $I_0$ $(m^{-2}s^{-1}sr^{-1})$ |
|---|---|---|---|---|
| Greisen [80,90] | 54 | 259 | 2.1 | $82.0 \pm 10.0$ |
| Crookes and Rastin [91] | 53 | 40 | $2.16 \pm 0.01$ | $91.3 \pm 1.2$ |
| Judge and Nash [76] | 53 | SL | $1.96 \pm 0.22$ | - |
| Dragić et al. [17] | 44.85 | 78 | - | $84 \pm 4$ |
| Briki et al. [21] | 35.76 | 38 | $1.82 \pm 0.11$ | $68.77 \pm 1.94$ |
| Arneodo et al. [93] | 24.54 | SL | $1.91 \pm 0.10$ (stat) $\pm 0.15$ (syst) | $75.4 \pm 1.3$ (stat) $\pm 1.5$ (syst) |
| Fukui et al. [81] | 24 | SL | - | $73.5 \pm 2.0$ |
| Sogarwal et al. [92] | 19 | SL | $2.10 \pm 0.25$ | $66.70 \pm 1.54$ |
| Karmakar et al. [94] | 16 | 122 | 2.2 | $89.9 \pm 0.5$ |
| Bhattacharyya [95] | 12 | 24 | $1.85 \pm 0.10$ | - |
| S. Pal [77] | 10.61 | SL | $2.15 \pm 0.01$ | $62.17 \pm 0.05$ |
| Pethuraj et al. [83] | 1.44 | 160 | $2.00 \pm 0.04$ (stat) $\pm 0.16$ (syst) | $70.07 \pm 0.02$ (stat) $\pm 5.26$ (syst) |
| Present data | 4.60 | 2657 | $2.145 \pm 0.046$ | $127.7 \pm 2.2$ |

The measurements reported here become an important input for future muon imaging projects at high altitudes and low latitudes. These high altitudes are benefited from the increased muon flux rate observed. For future muon imaging of the Monserrate Hill in Bogotá, the detectors will be upgraded to perpendicular planes of triangular scintillator strips readout by Silicon photomultipliers, with the use of wavelength-shifting fibers, in that way muon trajectory information can be obtained with high spatial resolution. Our experience with this initial study led us to conclude that it is better to have the detectors within a sealed space to fully protect them from wind, rain, and other climate conditions that could deteriorate their performance introducing additional systematic uncertainties.

**Author Contributions:** Conceptualization, C.Á.; methodology, C.B., C.Á. and G.R.; software, C.B.; validation, C.B., C.Á. and G.R.; formal analysis, C.B. and C.Á.; investigation, C.B., C.Á. and M.S.; resources, C.Á.; data curation, C.B.; writing—original draft preparation, C.B. and M.S.; writing—review and editing, C.B., C.Á. and M.S.; visualization, C.B. and M.S.; supervision, C.Á.; project administration, C.Á.; funding acquisition, C.Á. All authors have read and agreed to the published version of the manuscript.

**Funding:** This research was funded by the ministry of science, technology, and innovation of Colombia (MINCIENCIAS), through research program 70141-2021, and by the faculty of science of Universidad de Los Andes, Colombia through project INV-2020-105-2072.

**Data Availability Statement:** Data is presented in Table 2 of the present article. The raw data are available on request from the corresponding authors.

**Acknowledgments:** We would like to thank the High Energy Physics Laboratory members and students for their contributions to the writing of this paper. Special thanks to Paula Guzmán for her great help during the data collection and calibration processes. We also thank Luis Carlos Gómez, Jhon Restrepo, and the Micromechanics Workshop crew for their help with the building, mounting, and maintenance of the hodoscope structure. Finally, we thank Jhony Turizo and the Electronics Workshop for the software development required for the data acquisition.

**Conflicts of Interest:** The authors declare no conflict of interest.

## Abbreviations

The following abbreviations are used in this manuscript:

| | |
|---|---|
| BCD | Binary Coded Decimal |
| BSM | Beyond Standard Model |
| CV | Cutoff Value |
| CR | Cosmic Rays |
| DAQ | Data Acquisition |
| HV | High-Voltage |
| LFV | Lepton Flavor Violation |
| LG | Light Guide |
| LLT | Low-Level Threshold |
| PMT | Photomultiplier Tube |
| PW | Pulse Width |
| QDC | Charge to Digital Converter |
| SL | Sea Level |
| SM | Standard Model |
| TDC | Time to Digital Converter |
| ToF | Time of Flight |

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
