# Peer review of "Atmospheric Muon Flux Measurement near Earth’s Equatorial Line"

_instruments, doi:10.3390/instruments6040078_

Round 1
Reviewer 1 Report
The submitted manuscript describes the actual status of an R&D work conducted for muon imaging of Monserrate Hill, Bogota, Colombia. Authors report their recent measurements of muon flux at high-altitudes near the equator. The manuscript is well written. Maybe it is better to reduce the description about accelerator-based researches. The description about muon imaging seems to be cherry picking, thus I suggest to expand this part with further references. The applied instruments and analysis methods are presented in sufficient details according to the standards of the journal. The presented data provides useful reference for future experiments. I suggest to accept the manuscript after minor revision.
My suggestions and questions are listed as follows.
- line 17: "Muons were discovered by Anderson and Neddermyer in 1936 by studying cosmic rays (CR)" -> "Muons were discovered by Anderson and Neddermyer in 1936 via studying cosmic rays (CR)"
- Intro, 2nd paragraph, lines 24-30: Could you please add at least one reference that discusses these topics? Maybe it is merit to mention that muon is one of the elementary particles of the Standard Model (SM) before you discuss physics beyond the SM.
- The lines 25-61 mainly focuses on accelerator based experimental works. Cosmic-ray physics experiments, both satellited-based (e.g, AMS2 https://ams02.space/publications) and ground-based (e.g. Pierre Auger Observatory https://www.auger.org/science/publications/journal-articles) experiments go beyond the energy of accelerator-based experiments with approx. 8 orders and improve our knowledge about cosmic-rays, such as precisional measurement of the spectra of primary cosmic-rays in space or evolution of extensive air showers in Earth's atmosphere. These experimental works, especially ground-based measurements seem to be closer to your research than accelerator-based research.
- lines 66-67? What kind of models? Please, add at least a reference to the models that aim to stimulate muon production.
- lines 88-95: I suggest to insert this paragraph into the introduction which focuses on mouns and particle physics.
- line 96 "Álvarez" -> "Alvarez et al."
- line 101: "measured vertical muons" -> "measured muons"
- line 101: "Álvarez" -> "Alvarez et al."
- Lines 110-123:
If my understanding is correct, Tanaka did not yet work with Nagamine in the early 1990s. Nagamine et al [27] measured the muon shadow of Mount Tskububa, but they did not resolved the internal structure of the mountain with muons. Tanaka et al (2007) applied emulsion detectors to measure crater of Mt. Asama after eruption 2004 and they found a high-density region due to deposit of magmatic materials. Later, Tanaka et al. utilized scintillator detectors to visualize first time the magma movements in the conduit of the erupting Satsuma-Iwojima in 2013.
Tanaka et al. High resolution imaging in the inhomogeneous crust with cosmic-ray muon radiography: The density structure below the volcanic crater floor of Mt. Asama, Japan https://www.sciencedirect.com/science/article/abs/pii/S0012821X07005638
Tanaka et al. Radiographic visualization of magma dynamics in an erupting volcano https://www.nature.com/articles/ncomms4381
- Lines 124-137: This paragraph aims to summarize the achievement in the last 10 years in field of volcano muography. Since this is the motivation of the work, I suggest to expand this paragraph with mentioning the below works and cite some further references.
- "One of the motivations for using muons is the increased resolution compared to traditional gravimetric methods [28]."
Muography is rather a complementary tool than a new tool instead of gravimetry. Indeed, it helps to improve the spatial resolution and combining muography and gravity allowed to reconstruct the 3D density structure of volcanic edifices for detecting magmatic bodies (Nishiyama et al, 2017) and fluids (Rosas-Carbajal et al, 2017).
Nishiyama et al. 3D Density Modeling with Gravity and Muon-Radiographic Observations in Showa-Shinzan Lava Dome, Usu, Japan https://doi.org/10.1007/s00024-016-1430-9
Rosas-Carbajal et al. Three-dimensional density structure of La Soufrière de Guadeloupe lava dome from simultaneous muon radiographies and gravity data https://agupubs.onlinelibrary.wiley.com/doi/full/10.1002/2017GL074285
- The DIAPHANE is one of the largest muography experiment that made significant contribution to volcano muography via monitoring of movement of subsurface volcanic fluids and their abrupt changes in combination with seismic measurements in La Soufriere volcano that provide useful information for assessment of hazard level (Le Gonidec et al, 2018).
Le Gonidec et al. Abrupt changes of hydrothermal activity in a lava dome detected by combined seismic and muon monitoring https://doi.org/10.1038/s41598-019-39606-3
- Another ongoing large experiment is the Sakurajima Muography Observatory (SMO) which is modular system built from gaseous tracking detectors (Olah et al, 2018) at Sakurajima volcano, Kyushu, Japan. SMO observed the formation of magmatic plug after deactivation of erupting crater (Olah et al, 2019) and it was applied for monitoring changes on the surface of volcanic edifice by mudflows and erosion (Olah et el, 2021).
Olah et al. High-definition and low-noise muography of the Sakurajima volcano with gaseous tracking detectors https://www.nature.com/articles/s41598-018-21423-9/
Olah et al. Plug Formation Imaged Beneath the Active Craters of Sakurajima Volcano With Muography https://agupubs.onlinelibrary.wiley.com/doi/10.1029/2019GL084784
Olah et al. Muographic monitoring of hydrogeomorphic changes induced by post-eruptive lahars and erosion of Sakurajima volcano https://www.nature.com/articles/s41598-021-96947-8
- The structure beneath the crater Stromboli volcano was also measured with muography using emulsion detectors by Tioukov et al (2019).
Tioukov et al. First muography of Stromboli volcano https://www.nature.com/articles/s41598-019-43131-8
- For the MEV experiment, I suggest to mention that it is conducted at high altitude (approx. 3 km). An earlier experiment by Carbone et al also present useful experiences that is connected to your work.
Carbone et al. An experiment of muon radiography at Mt Etna (Italy) https://doi.org/10.1093/gji/ggt403
- Lines 138-139: "Recent studies on muon tomography for archaeological internal structures scanning have been proposed in Mexico to study the Sun Pyramid in Teotihuacan [34,35]." This sentence rather focusing on archeology. I suggest to add this to the end of paragraph about archeological applications. It merit to mention that there are ongoing developments for 3D scanning of the pyramids (Bross et al, 2022.). Besides pyramids, Greek tumuli (Marteu et al, 2018) and Japanese kofuns are also investigated with muography.
Bross et al, “Tomographic Muon Imaging of the Great Pyramid of Giza”, Journal of Advanced Instrumentation in Science, vol. 2022, Mar. 2022. http://journals.andromedapublisher.com/index.php/JAIS/article/view/280/114
Marteu et al. DIAPHANE: muon tomography applied to volcanoes, civil engineering, archaelogy https://doi.org/10.1088/1748-0221/12/02/C02008
Tanaka et al. Muography as a new tool to study the historic earthquakes recorded in ancient burial mounds https://doi.org/10.5194/gi-9-357-2020
- lines 164-186: My impression is that it would be better to move these two paragraphs into the introduction where you present the muons.
- Section 4.3: I suggest to provide information about the energy threshold/cut of experimental apparatus.
- How efficiently do you suppress atmospheric electrons/protons in your tracking system? I suggest to mention it in the manuscript.
- line 451: I assume that the unit is missing after the value of vertical muon flux rate.
Author Response
We thank the comments and questions sent by the referee which have been very useful to improve the quality of the manuscript. Please find below specific answers to each comment/question.

Reviewer 2 Report
Dear Authors,
the paper draft is rather interesting, and presents a classical-style simple detector setup with just sufficient angular resolution for a relevant muon flux angular dependence. Despite the fact that the results are re measurement of known information, the detector system description is rather clear and will be an excellent basis for future measurements of higher complexity and archeological/geoscientific interest.
I find two main shortcomings of the paper. One is related to the introduction. It must be noted that the Introduction is rather rich in useful information, and clearly explains a number of muography aspects. However, it misses a considerable fraction of relevant publications. (Or in turn: the impressive 63-entry long reference list has quite some irrelevant entries). Below I have suggestions to improve, particularly by citing some of the recent and relevant publications. Some are in-line with other remarks, please kindly consider those as well.
The other issue, which is also easy to resolve, is a probable mistake related to the key conclusion of the paper, where "integrated flux" is integrated over energy in previous references, and integrated over flux in the present paper. If I am not mistaken, please fix this error. I ask the Authors to add some more references related to the flux angular dependence, as there are many (very many) over the past decades.
---------------
Interesting introduction, discussing many interesting aspects.
However, physics BSD seem pretty far from muography, therefore I would suggest to drastically reduce on that, and instead bring out the more classical aspecs: e.g. instead of the exotic Higgs to muon pair decay, one can mention the Nobel-prize winning 4-lepton decay. I suggest to boil it down to a more basic introduction, where the question is, why muons are so special (e.g. CMS has "M" it its name): leptons as fundamental particles, experimentally very high penetration length and long lifetime. Most readers may be less familiar with frontier particle physics than the Authors are.
Muography applications: the text is clear, well organized, and gives also a good introductory. However, there are a number of highly relevant recent activities are missing, which are comparable, if not more important, than those already mentioned:
- Diaphane: worlds most extensive study of volcanoes, understanding the hydrothermal activities
K. Jourde et al, Muon dynamic radiography of density changes induced by hydrothermal activity at the La Soufrière of Guadeloupe...
https://doi.org/10.1038/srep33406
- SMO: worlds largest existing detector system, and the only observatory aimed at an actually active volcano
L. Oláh et al, High-definition and low-noise muography of the Sakurajima volcano with gaseous tracking detectors, Sci. Rep. 8$
D. Varga et al, High Efficiency Gaseous Tracking Detector for Cosmic Muon Radiography. AHEP 2016, 1962317
- discovery of magma dynamics:
H. K. M. Tanaka et al, Radiographic visualization of magma dynamics in an erupting volcano. Nat. Commun. 5, 3381 (2014) https://doi.org/10.1038/ncomms4381
Nice reviews:
Tanaka, H.K.M, Olah, L., Varga, D. (eds) Muography: Exploring Earth's Subsurface with Elementary Particles,
ISBN: 978-1-119-72302-8 Wiley AGU Books, 2022.
Bonomi G. et al., 2020. Applications of cosmic-ray muons. Progress in Particle and Nuclear Physics 112, 103768. https://doi.org/10.1016/j.ppnp.2020.103768
Tanaka H.K.M. and Oláh L., 2018. Overview of muographers.
Phil. Trans. R. Soc. A. 377, 20180143. https://doi.org/10.1098/rsta.2018.0143
Leone G. et al., 2021 Muography as a new complementary tool in monitoring volcanic hazard: implications for early warning systems.
Proc. R. Soc. A., 477, 20210320 https://doi.org/10.1098/rspa.2021.0320
Muon flux general comment: it is true that muon flux on surface follows the cos-squared law. However, there are important details related to this fact:
- the cos-squared law fails very strongly for high energy muons. Note that the number of high energy muons actually _increase_ at lower angles.
- low energy ("open sky") muons depend more on pressure, high altitude weather conditions, solar activities than higher energies. Consequently, flux measurements at, for example, 10-100m depth, has less time dependence than surface measurements. Further consequence is that the "opacity", that is the ratio of (muons through material)/(muons in open sky) is not a very precise quantity, due to the errors on the latter.
I suggest to re-formulate a bit the first part of "transmission muography" section, in the sense that one measures the _flux_ and then infers the density. (And this is why a high precision detector, such as that discussed in the present draft, is mandatory).
l. 242, 244: "The understanding of the atmospheric and environmental effects over muon flux intensity represents an advantage for the performance of attenuation muography. This allows estimating the time required to image certain targets."
These two sentences do not have anything to each other! Imaging time depends on detector size, angular resolution, depth, and required precision of density:
DOI:10.1098/rspa.2021.0320 (Quoted also above!)
--------
l.296. Please quote the angular resolution of the detector, which is around 50mrad (25cm/5m), that is around 3 degrees. Solid angle is only relevant for flux calculation.
---------
The measurement gives a nice classical efficiency measurement from stacking. What is the reason for C2 and C3 being around 98%? One would expect consistent efficiency above 99%. Can this be misalignment? Note that the QDC distributions are extremely clear on Figure 4.
How Figure 4 was obtained, is there an LLT imposed on the individual signals to obtain these distributions?
4.3 "Detector performances": the first two paragraph explains the HV setting by reaching the plateau. Please qantify these, and provide figures on:
- HV plateau measurements for the scintillators
- LLT scan as in the text ("plotted number of 4-fold coincidences")
Both these above are highly useful for experimentalists.
l.424: "For the zenith angle of 90 degrees, we expect zero muon flux, any measured value corresponds to the background". This is in fact far from truth, see particularly
https://www.nature.com/articles/s41598-018-21423-9
Horizontal muon flux is around 0.1/m2/s/sr (1/1000 of vertical).
Furthermore, background is extremely complicated (see e.g.
https://academic.oup.com/gji/article/206/2/1039/2606015
therefore this method: "accidental background is subtracted from the measurements performed at the other zenith angles" is not correct.
I think the measurements are just fine as they are, and I suggest not to subtract any "background". In the table 2, please quote this measured flux at 90 degree. Note that this flux is not the physical flux at 90 degree! The detector angular resolution is 3 degrees, and the flux near to horizon changes very rapidly with angle. Therefore, 3 degrees integrates over a strongly changing region. The detector geometry on the other hand is very well defined (square), so it is useful for any future data comparisons.
Question: would it not be preferable to use more standard SI units? Most muographers use meter and second units for flux.
----------------------
Discussions:
it is nice to see that indeed flux increases considerably when one goes to higher altitudes, practically doubles (100% increase) at the high altitudes of the present paper relative to SL.
I would like to stress that the measurements in the present paper are intersting and relevant, and worth publishing. However, citing 4 previous measurements is far-far from a reasonable literature search.
Among many-many which one finds with googleing the question out, there
is this nice measurement:
why not to cite it in the present draft. Note that in this paper there is a nice table of different measurements for comparison, so references can be taken from that. I suggest to look up the (maybe half a century?) literature on the same measurement, which was part of maybe most of all nuclear physicist students laboratory course, and cite some which seem relevant.
Important remark:
In Table 3, the VERTICAL INTEGRATED MUON FLUX is shown. Integrated flux means here integrated over energy, and not angle. Therefore, the present data, which is I_0=0.766, should be 12.76, right? The 25.5 seems the integrated over angle (which is too large by the factor 2pi/3).
Author Response

(The authors gave the same response as above.)

Reviewer 3 Report
The manuscript is worth for publication.
Author Response
We have performed various changes. We attached a document with the details.

Round 2
Reviewer 1 Report
Thank you for revising your manuscript. I suggest to the editors to accept your manuscript.
Author Response
We have performed a thorough revision of spelling and grammar errors, we list below the correction performed to the manuscript. Once again, thank you for all your suggestions.

Reviewer 2 Report
The improved draft is reasonable and useful for the community.
Author Response

(The authors gave the same response as above.)
